# Cognitive–Behavioral Profile in Pediatric Patients with Syndrome 5p-; Genotype–Phenotype Correlationships

**DOI:** 10.3390/genes14081628

**Published:** 2023-08-15

**Authors:** Cristina Bel-Fenellós, Chantal Biencinto-López, Belén Sáenz-Rico, Adolfo Hernández, Ana Karen Sandoval-Talamantes, Jair Tenorio-Castaño, Pablo Lapunzina, Julián Nevado

**Affiliations:** 1Departamento Investigación y Psicología en Educación, Facultad de Educación, Universidad Complutense, 28040 Madrid, Spain; mbel@ucm.es (C.B.-F.); alameda@edu.ucm.es (C.B.-L.); 2Departamento de Estudios Educativos, Facultad de Educación, Universidad Complutense de Madrid, 28040 Madrid, Spain; bsaenzri@edu.ucm.es; 3Departamento Economía Financiera y Actuarial y Estadística, Facultad de Comercio y Turismo, Universidad Complutense, 28040 Madrid, Spain; adolfher@ucm.es; 4Instituto de Genética Médica y Molecular (INGEMM)-IdiPAZ, Hospital Universitario La Paz, 28046 Madrid, Spain; dra_talamantes@hotmail.com (A.K.S.-T.); jaira.tenorio@salud.madrid.org (J.T.-C.); plapunzina@gmail.com (P.L.); 5Centro de Investigación Biomédica en Red de Enfermedades Raras (CIBERER), Instituto de Salud Carlos III (ISCIII), 28029 Madrid, Spain; 6ITHACA, European Reference Network, Hospital la Paz, 28046 Madrid, Spain

**Keywords:** behavioral assessment, cognitive assessment, Cri du Chat syndrome, 5p- Syndrome, SNP-arrays

## Abstract

(1) Background: 5p minus Syndrome (S5p-) is a neurodevelopmental disorder caused by a deletion in the short arm of chromosome 5. Among the phenotypic characteristics of S5p-, the most characteristic and representative element is a monochromatic cry with a high-pitched tone reminiscent of a cat’s meow. Individuals may also show great phenotypic heterogeneity and great genetic variability. Regarding cognitive–behavioral aspects of the syndrome, the studies are scarce and do not establish a general profile of the main cognitive–behavioral particularities that this syndrome presents. The main objective of this work was to describe the development profile of a cohort of 45 children with 5p minus Syndrome, concerning the biomedical, genetic, cognitive, and behavioral aspects. Establishing putative genotype–phenotype (cognitive–behavioral profiles) relationships in our cohort, from an interdisciplinary approach. (2) Methods: A selection of instruments of measures was selected for neuropsychological assessment (3) Results: In general, children with S5p- have a higher cognitive level than a communicative and motor level. Language difficulties, especially expressive ones, influence the frequency and severity of the most frequent behavioral problems in S5p. The most significant problem behavior of children with S5p-, especially girls, is self-harm. Compulsive behavior, limited preferences, and interest in monotony are significantly more frequent in subjects with better cognitive levels. We also find a significant correlation between the size of the loss of genetic material on 5p and the cognitive level of the subjects. (4) Conclusions: We described for the first time, the cognitive–behavioral profile of a cohort of minors with S5p-. Remarkably, it was found that language, especially of an expressive nature, modulates the most frequent behavioral aspects in subjects with lower cognitive levels, so it is essential to develop verbal or alternative communication strategies adjusted to these individuals.

## 1. Introduction

Cri du Chat Syndrome, also known as Chromosome 5p- Deletion Syndrome or 5p- Syndrome (hereinafter S5p-; OMIM #123450), is a neurodevelopmental disorder of genetic cause [1] due to a deletion in the short arm of chromosome 5. The size of the deletion varies from the total loss of the short arm to only the 5p15.3 region [2,3,4,5,6,7]. Among the phenotypic characteristics of S5p-, the most characteristic and representative element is that of a monochromatic cry with a high-pitched tone reminiscent of a cat’s meow. However, people affected by this syndrome have great phenotypic heterogeneity and great genetic variability [7], which is related to both the amount of genetic material lost and to the location of the deletion. Regarding cognitive–behavioral aspects, the studies are scarce, most of the existing ones are usually focused on specific aspects of development or the alteration that causes the deletion in the short arm of chromosome 5, and they do not establish a general profile of the main cognitive–behavioral particularities that this syndrome presents. Most of the authors cite profound intellectual impairment as one of the main features of S5p-. However, only Cornish and his team conducted an empirical study on the intellectual functioning of people affected by the syndrome [8]. They analyzed the scores of 26 subjects aged between 6 years and 4 months to 15 years and 5 months (mean 8 years and 3 months), on different measures: Intelligence (WISC-III, [9]), Vocabulary (British Picture Vocabulary Scales (BPVS) [10]), Grammar (Test of Reception of Grammar (TROG) [11]), Expressive Language (Expressive one Word Picture Vocabulary Test-R (EOWPVT) [12]), the Expression Subscale of the Reynell Language Development Scale [13]), and Articulation (Goldman–Fristoe Test of Articulation (AGLC), [14]). Among the different tests applied and regarding cognitive aspects, it is remarkable that the children obtained an IQ average score of 47.81. On the verbal scale, the mean obtained score was 50.3. They found no significant differences between the verbal and manipulative scales. Cornish and colleagues also showed a higher prevalence of people with moderate than severe intellectual disability. These results coincide with data collected in other previous studies [15].

On the other hand, Campbell evaluated 35 children and youth, aged three to eighteen, with the Battelle Developmental Inventory [16]. The published work confirmed the hypothesis that reflects a significant cognitive delay in the subjects of the sample. The mean age of the children and young people evaluated was 108 months, while the average age of general development was less than 35 months. However, there are also published works in which the subjects analyzed do not present cognitive delay or intellectual disability. Interestingly, all of them had small deletions in the distal area of the short arm of chromosome 5 [2,17,18,19], or showed an interstitial deletion [20,21].

In terms of behavioral aspects, individuals with S5p- are very curious about novelty, and interested in what is happening around them and others. The most relevant symptoms of their character and behavior are having a marked sense of humor, being affectionate, scary, and shy. They rarely present withdrawal or psychotic behavior as reported by Dikens and Clarke [22] using the Aberrant Behaviour Checklist [23]. Although different works and investigations have highlighted that people with S5p- present important and varied behavioral problems [15,22,24,25,26,27,28,29,30], most of these studies have focused on attention and hyperactivity problems, present in more than half of the children evaluated, as reported by Wilkins et al. [31], using the Vineland Social Maturity Scale [32], but there is little research on self- and hetero-aggressive behaviors [16].

In general, no correlation between age and aggressive behaviors was found, but a high correlation between self-injurious behaviors and stereotyped behavior can be observed. As in other genetic syndromes, some of the behavioral symptoms described of S5p- people overlap with the symptoms of Autism Spectrum Disorder (ASD): limited interests, repetitive behavior or stereotypies, etc. In most cases, the nature of this association between genetic syndromes and ASD is still unclear [33], and there seems to be an ASD profile associated with these disorders, different from that of idiopathic ASD [26]. Regarding S5p-, research has shown that ASD symptoms in people with S5p- are less severe than those in people with the same cognitive ability and that there is a higher prevalence among subjects with more sleep problems and higher levels of fatigue [34,35]. Finally, concerning other genetic syndromes, the S5p- people have fewer features of ASD than people with Angelman Syndrome or Cornelia de Lange Syndrome (SCdL) [36,37,38].

The general objective of this work is to establish the relationship between genetic aspects, biomedical aspects, and the cognitive–behavioral profile of the children in the sample, from an interdisciplinary approach. Trying to identify putative reference lines for educational care in children of pediatric age with 5p- Syndrome.

## 2. Materials and Methods

### 2.1. The Cohort

The final sample was composed of 45 children, 13 men, and 32 women. The age of the subjects was between 7 months, and 13 years and 1 month (mean age 6 years and 10 months). Most belonged to the 5p- Syndrome Foundation and a small percentage, 2 girls, came from the database of the Institute of Medical and Molecular Genetics (INGEMM) of the La Paz University Hospital in Madrid. All of them had been diagnosed with 5p- Syndrome previously, and their parents or legal guardians signed the informed consent from the Institutional Review Board of our Hospital and approved the study (HULP, Madrid, Spain) to participate in the research.

### 2.2. The Procedure

The procedure that followed in the collection of data began by contacting the families. They were explained what the research consisted of, and they were asked to sign the informed consent and to provide all the medical and psycho-pedagogical reports they had of their son or daughter. First, all the reports and graphic material provided were reviewed and a file was prepared with the collected data from each minor.

The neuropsychological assessment was carried out in two sessions in which the examiner administered the Battelle Developmental Inventory. Subsequently, in the interview with the family, the medical information collected from the reports was completed and the behavioral questionnaires with the parents were completed. In this way, the understanding of the behaviors evaluated and the coherence in the responses were guaranteed. It is important to note that the examiner (the psychologist, CBF), already knew most of the minors beforehand, having participated in and collaborated on the activities of the 5p- Syndrome Foundation for years. This facilitated the evaluations being carried out. To obtain the genetic data, it was necessary to perform microarray-SNP on all those subjects who had not been diagnosed by it, in the INGEMM of the Hospital La Paz. To do this, blood was drawn from the minors and analyzed in the manner described previously [7]. Once all the data were collected, the databases were elaborated on how to carry out the different statistical analyses. These were descriptive analyses, comparison analyses based on the sex variable and the existence of a second genetic alteration, correlation analyses, and multiple regression analyses.

### 2.3. SNP-Arrays

A genome-wide scan of 850,000 tag SNPs (Illumina Infinium CytoSNP-850k BeadChip) in most of the patients at INGEMM. They were analyzed by using the Chromosome Viewer tool contained in the Genome Studio package (Illumina, San Diego, CA, USA) using the manufacturer’s instructions. All genomic positions were based upon NCBI Build 37 (dbSNP version 130), and genomic coordinates were established according to the 2009 human genome build 19 (GRCh37/NCBI build 37.1). Deletion sizes were plotted on the genome browser using the University of California at Santa Cruz Genome Browser (http://genome.ucsc.edu/, accesed on 20 June 2022).

### 2.4. Neuropsychological Assessment Instruments

The selection of instruments to measure cognitive and behavioral aspects was made taking into account: (i) The characteristics of the participating subjects. The age of the minors, the presence of intellectual disability, and language and motor difficulties. (ii) The instruments used in published research, both with subjects with S5p- and with people with ID and characteristics similar to our participants. (iii) The evaluation procedure. Noting that the difficulties derived from evaluating subjects with intellectual disabilities mean that, in many cases, this evaluation must occur out of range or tests are used whose scales and standardized scores do not adjust to the population studied.

Several instruments collected information from the family since they present the format of questionnaires, and one was applied by the author of this work and completed with the information obtained in the informative interview. These aspects, together with the difficulties in assessing people with ID and great support needs, and the scarcity of specific and/or scaled instruments for this group, allowed us to establish that the instruments were as follows.
Battelle Development Inventory [39]. The Battelle Developmental Inventory is a battery designed to assess the key developmental skills of children from birth to age 8. It was developed by Newborg, Stock, and Wnek in 1984 and translated and adapted into Spanish in 1996 [39]. Its application is individual and typified. It consists of 341 items grouped into five areas: Personal/Social, Adaptive, Motor, Communication, and Cognitive. The collection of information is carried out through three procedures: (i) Structured examination. In which the examiner applies the items and provides the stimuli in a controlled environment. (ii) Observation. The observation of the subject in different environments, such as family and school, allows obtaining information regarding many of the aspects to be evaluated, especially those related to relationships and social interaction. (iii) Informative interview. The aspects and/or behaviors that can be evaluated more accurately with information provided by the family, teachers, or other people in their immediate environment, are asked and shared in an informative interview. By collecting information from three different sources, the information is contrasted and adjusted to the actual level of the subjects. Although some of the subjects in the sample were over the chronological age of the one collected in this test, none of them had a higher developmental age.Inventory of Behavioral Problems (BPI-01) [40]. It was developed by Rojahn in 2001, and translated and adapted into Spanish in 2008 by García-Villamisar. The Conduct Problems Inventory (BPI-01) is a questionnaire consisting of 52 items, which measures the self-injurious behavior (14 items), stereotyped behavior (24 items), and aggressive/destructive behavior (11 items) of the subjects evaluated. In addition, it contains 3 items, one in each scale, where you can add behaviors that are not explicitly listed in that category. Self-injurious behaviors are those that cause harm to one’s subject, stereotyped behaviors are inappropriate acts that occur habitually and repetitively, and aggressive or destructive behaviors are deliberate attacks against other individuals or objects. It is completed by the parents, or a person close to the subject, who must assess the frequency and severity of the behaviors described. The evaluation is made through a Likert scale of 4 points for the frequency of behavior (every month, every week, daily, every hour), and 3 points for severity (mild, moderate, severe). If the behavior does not occur, 0 is scored. Confirmatory factor analyses have provided support for the factorial validity of the measure. In the analysis of the internal consistency of BPI-01, they found a Cronbach’s α of 0.83. The subscales obtained alphas of 0.61 (self-injurious behavior), 0.79 (stereotyped behavior), and 0.82 (aggressive/destructive behavior) [40].The Repetitive Behaviors Questionnaire (RBQ) [41]. The RBQ is a questionnaire that assesses the frequency of 19 repetitive behaviors of children and adults with and without language. Respondents rate the frequency of operationally defined behaviors during the previous month. The response format is a Likert scale from 0 to 4 (never, once a month, once a week, once a day, or more than once a day). The results are grouped into 5 subscales: stereotyped behavior, compulsive behavior, limited preferences, repetitive speech, and insistence on monotony. It offers a clinical cut-off point for the different elements of the subscales. Behaviors that occur “once a day” or “more than once a day” were considered clinically important, that is, if a score of three points or more is obtained in a behavior. The stereotyped behavior subscale is composed of 3 items and evaluates the repetitive, and purposeless, movements of the body, a part of it, or objects. Compulsive behavior includes cleaning behaviors, hoarding, and unusual rituals and actions. This subarea is composed of 8 items. Limited preferences evaluate exaggerated attachment to people and objects, and consist of 3 items. The repetitive speech subarea includes 3 items and reflects whether the subject repeats phrases, words, questions, or what he has just heard; and the insistence on monotony values, with the preference for routine and order, consists of 2 items. In the analysis of internal consistency, they found Cronbach’s α of 0.80 for the total test and 0.70 for the subscales of repetitive behavior and stereotyped behavior. The alphas of the other three subscales were lower, limited preferences (α = 0.50), repetitive speech (α = 0.54), and insistence on monotony (α = 0.65) [41].Diagnostic Evaluation for the Severely Disabled (DASH-II) [42]. The Diagnostic Evaluation for the Severely Disabled (DASH-II) is composed of 84 items that allow the detection of psychiatric and emotional disorders in adults with ID and great support needs. It was developed by Matson in 1995, and translated and adapted into Spanish in 1999 by Novell, Forgas, and Medinyá. The DASH-II is the test that has the greatest international recognition to evaluate psychiatric problems in people with intellectual disabilities. In addition, we have its translation and adaptation into Spanish. Although the subjects of the sample were children, the lack of tests aimed at this specific population and the need to assess these aspects meant that it was included in the study and that the results were analyzed descriptively.

The instrument is divided into 13 subscales: impulse control problems, organic syndromes, anxiety, mood disorder, manic disorder, autism, schizophrenia, stereotypies, self-injurious behavior, elimination disorder, eating disorder, sleep disorder, and sexual behavior disorder. The informant must know the person evaluated for at least six months and scores the frequency, duration, and severity of the behaviors described in each item.

In this research work, we decided to analyze the scores obtained from the frequency and severity of the items evaluated, since the analysis of the duration of the period in which these behaviors are occurring is not significant, due to the young age of the participants. The results of the subarea—excretion disorders—(since most children do not control sphincters), and those of eating disorders and sexual behavior disorders (little significant due to the age and characteristics of the subjects) were also not included.

### 2.5. Statistical Analysis

Statistical analysis was performed with SPSS version 25 (IBM Corporation, Armonk, NY, USA). Descriptive analysis included mean ± SD for continuous variables and frequency tables for categorical variables. These categorical variables were expressed as 1 or 0, indeed grouped as “ever” having a given condition compared to “never” having the condition, taken from the two questionnaires and curated from medical records. Correlation associations were calculated using Pearson’s linear correlation coefficient (continuous variables) or Spearman’s Rho and Kendall’s tau_b (categorical variables). Comparisons between two groups (as based on sex or to have additional rearrangements) were performed either by Student’s *t*-test (for continuous variables) or by chi-square tests (for categorical ones). For more than two groups, ANOVA analysis (and Bonferroni’s post hoc tests) was run for continuous variables and z-tests between column proportions for categorical variables. PCA (principal component analysis) was used to validate our GFAP construct, containing Kaiser–Meyer–Olkin’s measure and Barlett’s test. Ward’s minimum variance method was the criterion used in hierarchical cluster analysis, and the number of clusters was selected using the Bayesian information criterion (BIC) or Akaike information criterion (AIC). A P-value (observed significance level) lower than 0.05 or 0.01 was considered to indicate a statistically significant or very significant difference, respectively. All statistical analyses were performed using the IBM SPSS statistical package, version 25.0, to carry out the descriptive analyses and to be able to elaborate the profile of the development of the minors of the sample concerning the biomedical, genetic, cognitive, and behavioral aspects. The first objective of our work was to make contingency tables with the different variables, and the frequencies and percentages were then established. To analyze whether, both in the total sample and the two cohorts studied, the variables included in the research present differences depending on the sex variable or the existence of a second genetic alteration, Student-*t*’s and Chi-square tests were performed. The correlation analyses were carried out with the Pearson correlation coefficient, and the multiple regression analyses were performed for the calculation of the predictive models, with the stepwise method. Before all statistical analyses, compliance with the assumptions required to perform them was verified. A significance level of 0.05 or less was established.

### 2.6. Limitations

The main limitation of this study can be denoted in three aspects: the sample size (it is a rare disease), the inability to have different tests adapted to patients with ID, and the restricted methods to collect the information from the patients.

## 3. Results

### 3.1. The Cohort

Detailed socio-demographic and genetic characteristics of this pediatric population with S5p- were previously included in Nevado et al., 2021 [7]. Briefly, the mean age of the subjects is 6 years and 10 months. Regarding the distribution by sex: men and women (1:2.46); that is, 29% men (13) and 71% women (34). The distribution of subjects by age and sex is shown in Appendix A. As can be seen, the largest number of individuals with S5p- is between 1 and 4 years (44.4%) and 8 and 10 years (33.7%). Regarding the demographic distribution of our sample, Figure 1 reflects the origin of the minors, and involved representation from almost all regions of Spain.

Two girls come from other countries, the United Kingdom, from a Colombian mother, and another from Venezuela.

The age at which subjects were diagnosed with S5p- varies from the first month of life to 12 years and 8 months. Four girls were diagnosed before the first month of life, one of them by prenatal diagnosis, and 70% of the subjects were diagnosed between the first and third month of life. Appendix A shows the neonatal data of this cohort. The mean gestational age of children is 38 weeks (SD = 2.64). Seven of them were born after 40 weeks of gestation, and eleven before gestational-week 37. None below 31. 59% (26 subjects) were born between weeks 37 and 40. The mean birth weight is 2526 g (SD = 666.8), which corresponds to the average weight of a neonate of 35–36 weeks, and the average length of 46 cm (SD = 3.3). Finally, the mean head circumference at birth is 31.9 cm (SD = 2.21). The main causes to perform genetic studies were prematurity, low weight, and suspected chromosomal abnormality. During the first months of life, eleven subjects had suction difficulties. Dysmorphic features and co-morbidities are described in Appendix A. Finally, the data of genomic rearrangements are presented in Appendix A. Briefly, deletions are encompassed between 4.35 and 36.60 Mb within 5p15.33–5p13.2 chromosomal bands and include different types of rearrangements (Table 1).

### 3.2. Cognitive Aspects

Table 2 shows the mental age (in months) obtained in the different areas of the Battelle Development Inventory, and the minimum and maximum scores recorded in each of them. Globally, their mental age is around 1.72 years old with an SD of 1.32 years, whereas the mean chronological age is around 6 years. Children obtain a higher average age in the cognitive area (24.16 months), followed by the personal area (22.89 months) and adaptive area (21.20 months). Regarding the subareas that make up the global language and motor score, the lowest level of development is achieved in the expressive language (16.20 months) and gross motor skills (18.09 months) subareas.

To delve into the most relevant cognitive aspects of the population with S5p-, the levels obtained in the motor and language areas were also analyzed.

#### 3.2.1. Motor Aspects

The mean age of all motor aspects of the sample is 18 months. The distribution of minors in the different motor milestones is shown in Appendix A. Under the age of 3 years, only one girl sits unsupported, and no subject walks or stands with assistance. Between 3 and 6 years, three children do not sit without support, and above 3 years, five children do not walk autonomously. Regarding fine motor skills, two minors are not able to grasp an object with their fingers and palm. Of the thirty-seven subjects older than 3 years, twenty can build a tower of two cubes, and, less than half, fifteen, copy simple strokes. As can be seen in Figure 2, the gross and fine motor level is similar in subjects who do not reach motor ages greater than 10 months, and in those who are above 50 months. In the rest of the minors, the level reached in the two subareas is discordant, with a greater number of subjects obtaining better scores in fine motor skills.

#### 3.2.2. Language–Communicative Aspects

The mean age of the total sample is 6 years and 4 months, and that of the language area is 1 year and 5 months. Being composed of children from the second trimester of life, it was considered necessary to divide the sample into two groups, taking into account the milestones of language acquisition in childhood. The first subgroup includes subjects from 0 to 3 years and, in the second, those older than 3 years. In this way, the communicative aspects are analyzed. We highlighted that, according to the normotypical development of children, the second group should have initiated oral communication. The mean age in the language area of subjects under 3 years is 5 months and that of those over 3 years is 1 year and 5 months. The mean ages of the two groups in the different language areas are shown in Table 3.

The increase in age corresponds to a slight increase in communicative resources. None of the children under 3 years of age reaches a linguistic age higher than 9 months of age. Nineteen children use natural gestures or hand signs and pictogram systems to support communication. Six have a literacy level corresponding to the first cycle of primary school.

Regarding the group over 3 years of age, Table 4 shows the distribution of subjects according to the linguistic age reached in the Battelle Development Inventory, and the age of the subareas Comprehension (receptive language) and Expression (expressive language). It also reflects the chronological age range of the subjects who obtain such scores. Almost half of the subjects obtain scores corresponding to the age of 0 to 12 months. No child reaches a linguistic age of 5 years. The highest score achieved in this area corresponds to a linguistic age of 52 months. The distribution is not uniform, nor does it remain constant in the different areas. Several subjects obtain, in different areas, scores corresponding to different age periods.

We also compare the scores obtained by the cohort in the two subareas of language, Comprehension (receptive language) and Expression (expressive language), which are reflected in Figure 3. We see that their distribution varies with the linguistic age of the individuals. There is evidence of a similarity of 0 to 10 months, being a lower expressive language level than the comprehensive one. Between 10 and 26 months of linguistic age, subjects present greater differences between areas, maintaining a better level of understanding than expression. However, subjects whose linguistic age is between 26 and 36 months maintain that greater difference between areas, but not all obtain better results in comprehension than in expression. Finally, with a linguistic age greater than 36 months, they present similar results in the two subareas.

### 3.3. Behavioral Aspects

Table 5 presents the data obtained from the behavioral rating scale of the BPI-01. Forty subjects scored on one of the items of the three subscales that make up the BPI-01: self-harm behavior; stereotyped behavior; and aggressive/destructive behavior, with the most frequent behaviors being self-harm (N = 37), with the highest mean score (4.60; SD = 3.79).

Table 6 presents the results of the RBQ test. The five subscales of the test, stereotyped behavior, compulsive behavior, limited preferences, repetitive speech, and insistence on monotony, collect the frequency of the behavior described in the last month. Thirty-nine children scored on one of the items on the scale, mainly on limited preferences (N = 21) and insistence on monotony (N = 18).

Finally, Table 7 shows the results of the Diagnostic Evaluation for the Severely Disabled (DASH-II). Forty-four subjects scored in any of the subscales used in the test: impulses, organic, anxiety, mood, mania, autism spectrum disorder, schizophrenia, stereotypies, self-injurious behavior, and sleep problems. However, none reaches the necessary score to be able to establish any of the diagnoses of the scale, whose score, as we have seen, is obtained through a Likert-type scale. The breadth of the ranges of scores achieved in each of the subscales is small, between 0–1 (organic scale/severity) and 0–7 (autism scales). Thirty-six of the children have some of the traits that the scale recognizes as characteristic of autism, and thirty-four, self-injurious behaviors. Stereotyped behavior (N = 31), sleep problems (N = 28), and impulsive behavior (N = 24) are also common. There are some scores in organic disorder (N = 3), mood disorder (N = 16), and mania (N = 20), but none in anxiety or schizophrenia.

### 3.4. Distribution of Cognitive Aspects according to Gender

We performed the Student *t*-test to analyze the distribution of cognitive variables according to sex. No significant differences were found in any of the areas evaluated: personal (APERS) (t = −0.004; *p* > 0.05), adaptive (AADAP) (t = 0.100; *p* > 0.05), gross motor skills (AMOTGR) (t = 0.047; *p* > 0.05), fine motor skills (AMOTFI) (t = −0.042; *p* > 0.05), total motor area (AMOTTOT) (t = 0.012; *p* > 0.05), receptive language (ALENGR) (t = −0.355; *p* > 0.05), expressive language (ALENGE) (t = 0.037; *p* > 0.05), total language area (ALENGT) (t = −0.118; *p* > 0.05), and cognitive area (ACOG) (t = 0.745; *p* > 0.05), nor in the total Battelle score (TOTAL) (t = 0.120; *p* > 0.05).

### 3.5. Distribution of Behavioral Aspects according to Gender

Table 8 and Table 9 show the statistics and results obtained in the different behavioral scales, after applying the Student’s *t* test. No significant differences were found according to sex in most of the subareas that compose them. Table 8 shows the statistics and results obtained in the subscales and totals of the BPI-01 test. Significant differences were found according to sex in the aggressiveness-severity subscale (AGRGR) (t = −2.504; *p* = 0.016), with women obtaining the highest score.

No significant differences were found according to sex in any of the aspects analyzed using the RBQ test.

Table 9 shows the statistics and results obtained in the subscales and totals of the DASH-II test. As mentioned, when describing the results, the size of the ranges of scores in this test was very low, so most of the means are close to zero. Significant differences were found according to sex in the mania-frequency (MANFR) subscale (t = −2.735; *p* = 0.009), sleep problems—frequency (PSUFR) (t = −2.389; *p* = 0.021), and sleep problems—severity (PSUGR) (t = −2.409; *p* = 0.020), all of which were more frequent among women.

### 3.6. Correlation between Loss of Genetic Material and Cognitive Aspects

One of the main hypotheses of our study was to find a significant correlation between the loss of genetic material in subjects with S5p- and their results in cognitive tests. To verify this, a correlation analysis was carried out using Pearson’s. As can be seen in Table 10, all areas of the Battelle Development Inventory, and the total score, correlate significantly with the size of the loss of genetic material. In all cases, a greater loss of genetic material means a lower score in all areas of cognitive development.

### 3.7. Correlation between Loss of Genetic Material and Behavioral Aspects

Pearson’s correlation analysis was also performed between the size of the loss of genetic material and the different behavioral scales applied. Table 11 shows that, in the BPI-01 test, the size of the loss correlates significantly with the frequency and severity of self-harm behaviors, and with test totals.

In the RBQ test, the size of the loss correlates significantly with compulsive behavior (CCOM) (r = −0.305; *p* = 0.042), which is lower when the loss of genetic material is greater, and in the Diagnostic Evaluation for Severely Disabled (DASH-II), with the frequency of mania behavior (MANFR) (r = 0.412; *p* = 0.005), the frequency of sleep problems (PSUFR) (r = 0.337; *p* = 0.025), and frequency and severity of stereotyped behavior (ESTFR, r = 0.331; *p* = 0.026; ESTGR, r = 0.299; *p* = 0.046) and test totals (TOTFR, r = 0.390; *p* = 0.008; TOTGR, r = 0.324; *p* = 0.030).

Figure 4 summarizes the correlates of loss of genetic material and cognitive–behavioral aspects.

### 3.8. Correlation between Cognitive Variables and Behavioral Aspects

To verify the relationship between cognitive variables and behavioral variables, a Pearson correlation analysis was performed between the cognitive results provided by the different areas of the Battelle Developmental Inventory and the applied behavior scales: the Behavior Problems Inventory (BPI-01), the Repetitive Behavior Questionnaire (RBQ), and the Diagnostic Evaluation for Severely Disabled (DASH-II) (see Appendix A). Regarding the BPI-01 test (Figure 5a), the frequency of stereotyped behaviors correlates significantly with all areas of the Battelle Development Inventory: personal (PERS), adaptive (ADAP), gross motor (MTGR), fine motor (MTF), total motor (MOT), receptive language (RECP), expressive language (EXP), total language (LENG), cognitive (COG), and total test (TOT). The severity of stereotyped behaviors correlates significantly with all areas, except for gross and fine motor ones. The frequency (AGRFR) and severity (AGRGR) of aggressive behaviors correlated significantly with the receptive, expressive and total language, and cognitive areas. AGRGR also correlates with the adaptive area. Total test frequency (TOTFR) and severity (TOTGR) scores correlate with the same areas: adaptive, receptive, expressive and total language, cognitive, and total test in frequency and severity.

Regarding the results of the analysis of the Pearson correlation between the cognitive variables and the RBQ test (Figure 5b), as in the previous test, stereotyped behavior correlates significantly with all cognitive areas. Compulsive behavior correlates significantly with all areas, except for expressive language. Limited preference correlates significantly with gross motor score and total motor score. Insistence on monotony correlates significantly with fine motor score and cognitive scores. All correlations of compulsive behavior, limited preferences, and insistence on monotony were positive, with a greater number of these behaviors in those subjects who have a higher cognitive level.

The results obtained from the correlational analysis of cognitive variables and the results of DASH-II (Figure 5c) show that the frequency and severity of stereotyped behavior also correlate significantly with all areas of the cognitive scale, as well as with sleep problem scores, both frequency, and severity. Correlations between the frequency and severity of total test scores were also significant with all cognitive areas, except for the total-severity score and fine motor area. Finally, mania frequency correlates significantly with receptive, expressive, and total language areas. Mania severity correlates with all areas, except for fine motor and personal ones.

## 4. Discussion

5p minus Syndrome (S5p-) is a low prevalence rare disease with a low number of previous investigations. A few groups of patients have been analyzed, but none of these have been cognitively assessed. There is only one published work [16] in which subjects have been evaluated with the Battelle Developmental Inventory, which was also used in our study. As in Campbell’s work, most of our subjects have significant cognitive delays. In this study, the authors evaluated American children and young people aged three to 18 years, with an average age of 108 months, which is a slight difference from our sample, whose average is 82 months. Our work is the first complete study of the Spanish population and the first study carried out with a sample composed entirely of minors, with the difficulty that this entails. The study focused on subjects of pediatric age, to establish early intervention models, since it is considered that they can be a determining factor in improving his/her quality of life.

Briefly, the nuclear characteristics of the S5p- subjects of pediatric age are microcephaly (91.1%), hypotonia (84.4%), hypertelorism (66.7%), and dental alterations (60%). The important characteristics that help us to describe the syndrome in this age, present between 60% and 50%, are the alterations of the auricular pavilions (57.8%), epicanthus (57.8%), wide nasal bridge (55.6%), visual alterations (55.5%), hearing disturbances (53.3%), micrognathia (51.1%), and respiratory problems (51.1%) (see Appendix A). The outstanding features serve as a basic reference guide for medical professionals and other specialists not familiar with the management of patients with S5p.

In both investigations (Campbell’s and ours), Battelle’s results of the personal area are better than those of the average score of general development (Total score), while the motor and language areas obtain lower results, with better scores in the first of these. We see that these traits are part of the profile, regardless of the context. In our cohort, the results of the cognitive area are the highest, and the total of the subjects presents a better cognitive level than language. As we have seen, this aspect is reflected in multiple previous works [8,43,44], which highlight this discrepancy for all cognitive levels. However, in our group, we highlighted that the eight subjects whose linguistic age is between 26 and 36 months maintain this difference between areas, but not all obtain better results in comprehension than in expression, and those with a linguistic age greater than 36 months (five subjects), present similar results in the two subareas. Although the number of subjects is limited, this could suggest that there are differences in the development of communicative aspects, depending on the mental age of the subjects.

Regarding motor development, of the 37 subjects older than 3 years, 32 walk autonomously; between 3 and 6 years, three do not; noting that between 3 and 4 years is the average age at which children with S5p- achieve it [45]. Also, according to the published data, they sit alone between 13 and 18 months, but none of our subjects of that age do, and they build towers of two cubes between 3 and 4 and a half years, although 17 of the subjects of our sample, of that age or older, do not.

The analysis by different methods of psychomotor work shows different results in the population of subjects with S5p- [46]. The result of the cognitive area is higher than the language and motor areas. This is especially interesting since reinforcing the linguistic and motor aspects will enhance cognitive resources. The behavioral profile of our children is similar to that described by [24] and is characterized by self-injurious behaviors, hetero-aggressive behaviors, limited preferences, and insistence on monotony, stereotyped behavior, sleep problems, and impulsive behavior. Self-injurious behaviors are more frequent than hetero-aggressive ones, contrary to what happened in the study by [15]. The results of the Conduct Problems Inventory (BPI-01) show that 82% of the subjects present self-injurious behavior; 73%, stereotyped behaviors; and 71%, aggressive behavior. These results are similar, although lower than those of the work of Collins and Cornish [26] who evaluated 66 children and young people with the same test and obtained higher percentages: 89%, self-injurious behavior; 82%, stereotyped behaviors; and 88%, aggressive behavior. In both studies, head-swinging blows to the head with different parts of the body, and self-biting stand out. In ours, in addition, the trichotillomania.

### 4.1. Behavioral Profile of Minors with S5p-

The analysis of the results obtained from the applied behavioral tests allows us to establish the behavioral profile of minors with S5p-, and to study if it is conditioned by the different moments of development of each of them. As highlighted in the description of the main characteristics of the population with S5p-, the behavioral profile is characterized by self-injurious behaviors, hetero-aggressive behaviors, limited preferences, insistence on monotony, stereotyped behavior, sleep problems, and impulsive behavior. Self-injurious behaviors are more frequent than hetero-aggressive behaviors and stereotyped behaviors include head swinging, blows to the head with different parts of the body, self-biting, and trichotillomania. The description in the Results exposes the relationship between the areas of cognitive development and the different behavioral aspects. It highlights that compulsive behavior, limited preferences, and interest in monotony are significantly more frequent in subjects with better cognitive levels. The last two characteristics are traits also present in the behavior of people with ASD. Therefore, children with S5p- and better cognitive levels may present a behavioral profile more similar to that of ASD people than those with lower cognitive levels. The rest of the aspects in which the relationship is significant, although more frequent in those with lower cognitive levels, correlates with the language area and/or some of its subareas (expressive or receptive). The area of expressive language is the one with the most correlations. It has already been highlighted that the cognitive level is usually higher than the linguistic level, so the lower the language level, the greater the hetero-aggressive, self-aggressive, and sleep problems. Multiple studies have highlighted the relationship between behavioral problems and communication difficulties [47]. The importance of early intervention for linguistic aspects as support for cognitive development has already been mentioned, but also, the development of communication strategies is essential to reduce behavioral problems. Therefore, it is essential to develop intervention models focused on the development of these strategies, and in many cases, they will have to introduce non-verbal communication systems. In this way, people with S5p- will be provided with a communication system from an early age, favoring the maximum development of their abilities and reducing the appearance of behavioral problems.

In the results of the BPI-01 test, Collins and Cornish [26] showed a significant correlation between self-injurious behavior and stereotyped and hetero-aggressive behavior, but not between stereotyped behavior and aggressive behavior. These authors also highlighted a significant negative correlation between aggressive behaviors and age; that is, as age increased, these behaviors decreased. In our study, all three behaviors have a significant correlation, and age does not correlate with any of them. As mentioned, there is an abbreviated version of the BPI-01 scale, the BPI-S. Its author performed the validation of this second scale by evaluating the behavioral problems of 1122 patients with intellectual disabilities (ID), aged 2 to 93 years, from different countries [48,49]. He compared them with the results that the subjects obtained in the long version, BPI-01, and correlated them with the data of another of the scales we used, the DASH-II. The diagnosis of the subjects was very diverse and served as a reference group to compare our sample with the population with ID. Normative data on people with ID from BPI-01 are grouped into age ranges. Our sample is included in the first two, 0–10 years, and 10.1–15 years. In our study, women have higher averages in all aspects analyzed by the test. The only aspect that is more frequent among children with S5p- than in the general group, is the frequency and severity of self-injurious behaviors of women. That is, girls with S5p- between 0 and 13 years old have a higher frequency and greater severity of self-injurious behaviors than girls with ID of the same age.

We highlight that despite the age difference between the two groups, children with S5p- score higher in stereotypies, autistic traits, self-injurious behavior, and sleep problems than the adult population with disabilities (compare to Rojahn et al. [48,49]). On the other hand, Moss et al. [37] analyzed the repetitive behavior of different genetic syndromes. They remarked that the S5p- profile only showed significant results in the object attachment category. In our sample, all scoring categories correlate significantly.

### 4.2. Cognitive–Behavioral Profile and Genotype of Minors with S5p-

As seen in previous works, most authors state that the severity of the phenotype and cognitive delay is associated with a greater loss of genetic material from chromosome 5 [8,50,51], although there are also studies in which this is not confirmed [52,53]. Our analyses were carried out to verify this hypothesis, by which genetic alterations may influence other biomedical, cognitive, and behavioral aspects of our children, and show that there are no significant correlations between the loss of genetic material and any of the neonatal data, except for those with suction difficulties, the presence of epicanthus, hypertelorism, or the presence of downward palpebral fissure or an ogival palate. All areas of the Battelle Development Inventory also correlate significantly.

In all these aspects, a greater loss of genetic material means a lower cognitive level, more suction difficulties, and a greater presence of the aforementioned physical features, except for the ogival palate, whose relationship is inverse. Indeed, a greater loss implies less presence of this alteration. The authors mentioned above did not assess the relationship of loss size with behavioral aspects. In our sample, this correlation is significant between the loss of genetic material, the frequency and severity of self-harm behaviors, and the total score of behavioral problems on the BPI-01 test; also, with the frequency of mania behavior and sleep problems, and the frequency and severity of stereotyped behavior and DASH-II test totals. In all of them, a greater loss of material from the short arm of chromosome 5 implies a greater frequency of these aspects. However, the size of the loss also correlates significantly with compulsive behavior (RBQ). Although this correlation is negative, therefore with smaller deletions, the greater the presence of compulsive behaviors in the subjects.

### 4.3. Sex as a Differentiating Factor

Among parents of children with S5p, there is a belief that sex is a determining factor in the cognitive and behavioral abilities of their children. The analyses carried out to verify whether sex can be a differential factor in the biomedical, cognitive, and behavioral profile of children with S5p- show that the appearance of a large mouth, respiratory problems, and visual alterations are more frequent among boys, and scoliosis among girls.

In our study, the severity of hetero-aggressive behaviors (BPI-01), sleep problems, and frequency of mania aspect (DASH-II) are more frequent among girls. However, Collins and Cornish [26] found no significant differences by sex in any of the aspects of the Conduct Problems Inventory (BPI-01). The results of our children in the Battelle Development Inventory show that sex is not determinant in any of its areas, unlike an American study in which the results of the cognitive area are significantly better in boys [16], so it can be said that there are discrepancies. However, it should be noted that the generic data and the differences between boys and girls in the US population are not known. In our sample, girls have a greater loss of genetic material (22.81 Mb) than boys (19.33 Mb), although this difference is not statistically significant.

## 5. Conclusions

We described for the first time, a cohort of individuals with S5p- exclusively of pediatric age, from an interdisciplinary point of view, systematically, and rigorously. Generally speaking, it is similar to other cohorts described, in terms of dysmorphic ad comorbidity features, but it is the first time that we know of, where a complete description of the cognitive–behavioral profile in minors has been obtained. No significant differences were found in the different variables depending on sex, despite the suggestion established by the parents and some previous studies.

We do find a significant correlation between the size of the loss of genetic material on 5p and the cognitive level of the subjects. In general, children with S5p- have a higher cognitive level than a communicative and motor level. This implies that these two aspects, corrected by age, must be two fundamental milestones in the first level of intervention. Compulsive behavior, limited preferences, and interest in monotony are significantly more frequent in subjects with better cognitive levels. Language difficulties, especially expressive, influence the frequency and severity of the most frequent behavioral problems in S5p-. Indeed, language, especially expressive language, modulates the most frequent behavioral aspects in subjects with lower cognitive levels, so it is essential to develop verbal or alternative communication strategies adjusted to these individuals. The most significant problem behavior of children with S5p-, especially girls, is self-harm.

## Figures and Tables

**Figure 1 genes-14-01628-f001:**
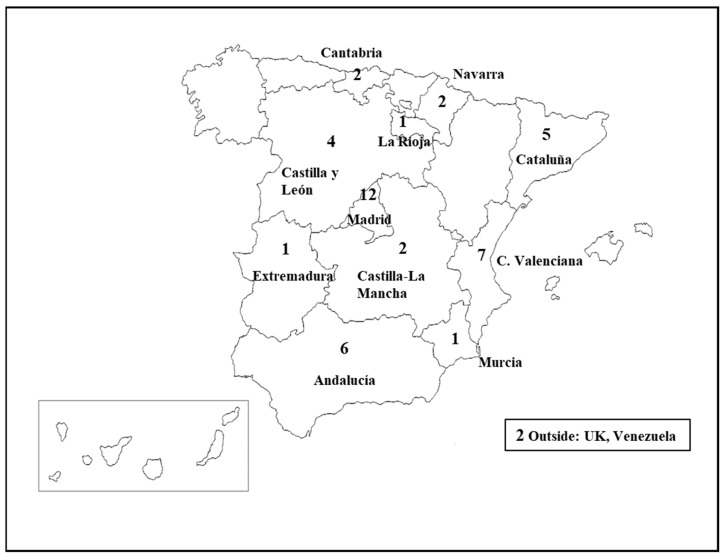
Distribution of the Cohort’s individuals. The numbers inside of the figure mean number of individuals by region.

**Figure 2 genes-14-01628-f002:**
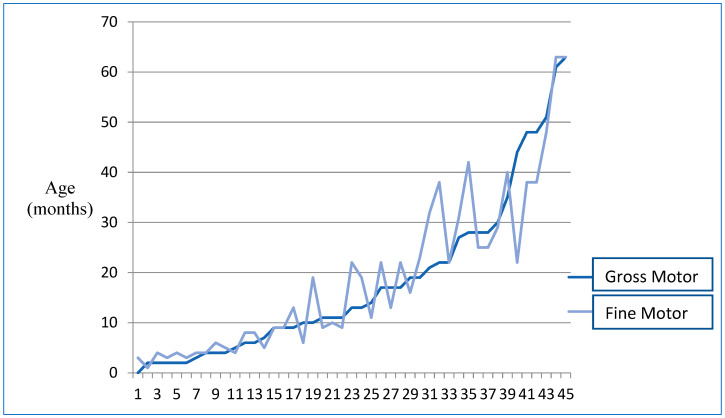
Distribution of Gross Motor Area and Fine Motor Area scores by using the Batelle development inventory.

**Figure 3 genes-14-01628-f003:**
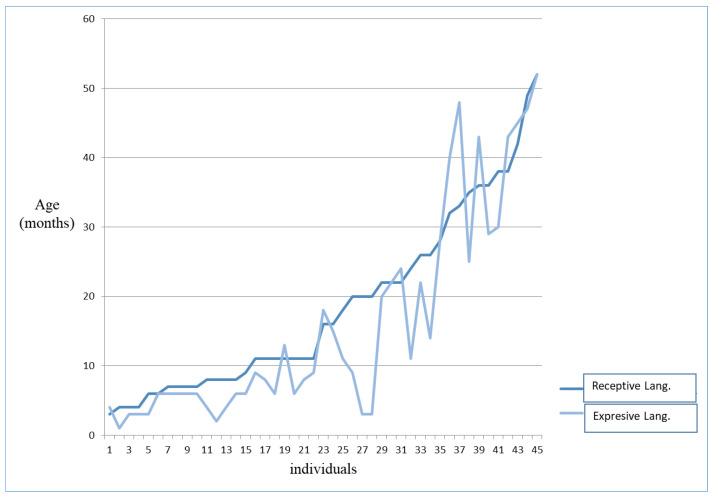
Distribution of Receptive and Expressive language area scores by using the Batelle development inventory.

**Figure 4 genes-14-01628-f004:**
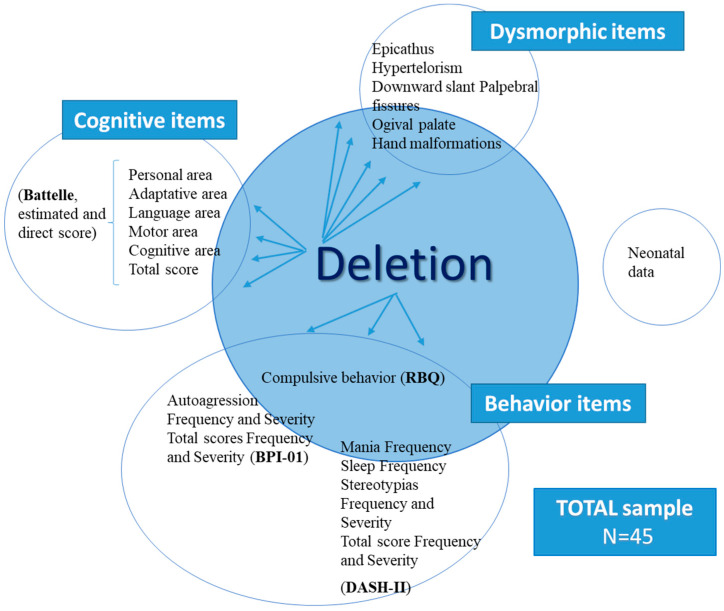
Schematic representation of the significant correlationships (*p* is significant < 0.05) between “loss of genetic material” and “cognitive–behavioral aspects”.

**Figure 5 genes-14-01628-f005:**
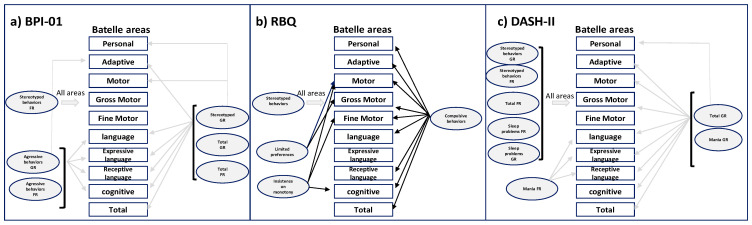
Schematic representation of correlates of behavioral problems (by BPI-01; RBQ, DASH-II tests) and cognitive scores (by Batelle development inventory) after applying Person correlation analysis. The arrow means correlations with a *p* significant < 0.05. Grey arrows; negative correlationships, black arrows; positive correlationships. FR, frequency; GR severity.

**Table 1 genes-14-01628-t001:** Type and frequency of rearrangements observed in the cohort.

Type	Frequency	%
Terminal deletion	43	95.50
Interstitial deletion	2	4.40
Familial Translocation	5	11.10
Translocation de novo	8	17.80
Terminal deletion originated from a parental mosaicism	1	2.00
Mosaicism	0	-
Ring chromosome	0	-
Additional duplication	20	44.40
Other rearrangements	3	6.70

**Table 2 genes-14-01628-t002:** Results of the areas of the Battelle Development Inventory.

Areas	Minimum Score	Maximum Score	Mean (Age in Months)	SD
Personal	2	64	22.89	16.45
Adaptive A	1	62	21.20	18.64
Motor	1	61	18.02	14.89
Gross Motor	0	63	18.09	16.43
Fine Motor	1	63	18.89	15.90
Language	3	74	17.51	15.65
Expressive Language	1	52	16.20	5.04
Receptive Language	3	74	18.27	12.59
Cognitive	2	90	24.16	19.88
Total	2	60	20.73	15.86

Note: Although some of the subjects in the sample were over the chronological age for the design of this test, none of them had a higher developmental age. In fact, all the individuals were two or more standard deviations below their chronological age. SD, standard deviation.

**Table 3 genes-14-01628-t003:** Results of the areas of the Battelle Development Inventory.

	Younger, <3 y (N = 8)	Older, >3 y (N = 37)
Mean age	1 y, 7 m	7 y, 4 m
Receptive Language	6 m	1 y, 7 m
Expressive Language	4 m	1 y, 5 m
Total language	5 m	1 y, 5 m

N, number of individuals; y, years old; m, months.

**Table 4 genes-14-01628-t004:** Distribution by linguistic age of the group over 3 years old.

Linguistic Age (Months)	Receptive Lang.		Expressive Lang.		Total	
	N	Chronologic Age	N	Chronologic Age	N	Chronologic Age
0–12	14	3–13	18	3–13.1	18	3–13
12–24	10	3.7–13.1	8	3.8–12.1	8	4.5–13.1
24–36	8	4.5–10.2	4	8.1–12.9	7	4.5–12.9
36–48	3	6.3–12.9	6	4.5–11	3	6.3–8.8
+48	2	8.7–8.8	1	8.7	1	8.7

N, number of individuals; Lang., language.

**Table 5 genes-14-01628-t005:** Descriptive analysis of the subscales and the BPI-01 test set.

		N	Range Scores	Mean Score (0–4 FR)+ (0–3 GR)	SD
Self-harm behavior	Frequency Severity	37 37	1–12 1–11	4.60 3.91	3.79 3.23
Stereotyped behavior	Frequency Severity	33 33	1–18 1–11	3.78 2.87	3.56 2.60
Destructive aggressive behavior	Frequency Severity	32 32	1–9 1–6	2.29 2.18	2.15 1.85
Total	Frequency Severity	40 40	1–26 1–23	10.67 8.96	8.05 6.54

Two types of scores were obtained. How often the behavior has appeared in the past 12 months (Frequency, FR), and how severe the behavior is (Severity, GR). Note: an individual without ID or behavioral problems does not score at all. N, number of patients who scored in the test; SD, standard deviation; BPI, Inventory of Behavioral Problems.

**Table 6 genes-14-01628-t006:** Descriptive analysis of the subscales and the RBQ test set.

	N	Range Scores	Mean Score (0–4)	SD
Stereotyped behavior	13	1–8	2.78	2.37
Compulsive behavior	8	1–6	0.38	1.72
Limited preferences	21	1–4	1.31	1.58
Repetitive speech	0	0	0	-
Insistence on monotony	18	1–4	0.98	1.30
Total	39	0–15	5.44	4.05

RBQ offers a clinical cut-off point for the different elements of the subscales. Thus, behaviors that occur “once a day” or “more than once a day” were considered clinically relevant (three or four points, respectively). An individual without ID nor behavioral problems does not score more than 1 point. N, number of patients who scored in the test; SD, standard deviation; RBQ, Conduct Problems Questionnaire.

**Table 7 genes-14-01628-t007:** Descriptive analysis of the subscales and the whole of the DASH-II scale.

		N	Range Scores	Mean Score	SD
Impulses	Frequency Severity	24 24	0–4 0–3	0.91 0.78	1.10 0.88
Organic	Frequency Severity	33 33	0–2 0–1	0.11 0.02	0.44 0.15
Anxiety	Frequency Severity	0 0	- -	- -	- -
Mood	Frequency Severity	16 16	0–4 0–2	0.78 0.47	1.13 0.69
Mania	Frequency Severity	20 20	0–4 0–2	1.02 1.02	1.20 1.20
Autism	Frequency Severity	36 36	0–7 0–7	3.22 2.58	2.25 1.95
Schizophrenia	Frequency Severity	0 0	- -	- -	- -
Stereotypies	Frequency Severity	31 31	0–6 0–4	2.09 1.58	1.70 1.41
Self-harm	Frequency Severity	34 34	0–7 0–6	2.53 2.09	2.06 1.76
Sleep Problems	Frequency Severity	28 28	0–4 0–4	1.43 0.98	1.28 0.98
Total	Frequency Severity	44 44	0–27 0–23	12.20 9.07	7.06 5.65

Two types of scores were obtained. How often the behavior has appeared in the past 12 months (Frequency) and how severe the behavior is (Severity). Psychopathology should be ruled out in scores below [42]: 8 points for impulses; 4–5 points for organic; 2 points for anxiety; 6 points for Mood aspects; 4–5 points in mania; 4–5 points in autism, 2–3 points in schizophrenia; 4–5 points in stereotypes; >1.5 points for self-harm injured or sleeping problems (based on our professional experience). N, number of patients who scored in the test; SD, standard deviation.

**Table 8 genes-14-01628-t008:** Statistics of comparison of the subscales and totals of the BPI-01 test.

	Men		Woman			
	Mean	SD	Mean	SD	t	*p* (Bilateral)
AUTFR	3.54	3.84	5.03	3.75	−1.20	0.24
AUTGR	2.69	2.98	4.41	3.24	−1.643	0.11
ESTFR	2.85	2.61	4.16	3.85	−1.12	0.27
ESTGR	2.23	1.878	3.13	2.83	−1.05	0.30
AGRFR	1.38	2.02	2.66	2.12	−1.85	0.07
AGRGR	1.15	1.63	2.59	1.79	−2.5	0.01 *
TOTFR	7.77	7.21	11.84	8.18	−1.56	0.13
TOTGR	6.08	5.52	10.13	6.63	−1.94	0.06

AUTFR = Self-harm behavior, frequency; AUTGR = Self-harm behavior, severity; ESTFR = Stereotyped behavior, frequency; ESTGR = Stereotyped behavior, severity; AGRFR = Aggressive behavior, frequency; AGRGR = Aggressive behavior, severity; TOTFR = Total frequency; TOTGR = Total severity; BPI, Inventory of Behavioral Problems. * *p* is significant < 0.05.

**Table 9 genes-14-01628-t009:** Statistics of comparison of the subscales and totals of the DASH-II test.

	Men		Woman			
	Mean	SD	Mean	SD	Mean	SD
IMPFR	0.77	1.09	0.97	1.12	−0.55	0.59
IMPGR	0.62	0.65	0.84	0.95	−0.79	0.43
ORGFR	--	--	0.16	0.52	−1.09	0.28
ORGGR	--	--	0.03	0.18	−0.63	0.53
ANSFR	--	--	--	--	--	--
ANSGR	--	--	--	--	--	--
HUMFR	0.31	0.75	0.97	1.20	−1.83	0.07
HUMGR	0.23	0.60	0.56	0.72	−1.47	0.15
MANFR	0.31	0.75	1.31	1.23	−2.74	0.01 *
MANGR	0.31	0.63	0.72	0.81	−1.63	0.11
AUTFR	3.54	2.37	3.10	2.23	0.60	0.56
AUTGR	2.85	2.19	2.47	1.87	0.59	0.56
ESQFR	--	--	--	--	--	--
ESQGR	--	--	--	--	--	--
ESTFR	1.77	1.70	2.22	1.72	−0.80	0.43
ESTGR	1.15	1.35	1.75	1.41	−1.30	0.20
AGRFR	2.31	2.16	2.63	2.11	−0.46	0.65
AGRGR	1.69	1.49	2.25	1.85	−0.97	0.34
PSUFR	077	1.01	1.72	1.28	−2.39	0.02 *
PSUGR	0.46	0.66	1.19	1.00	−2.41	0.02 *
TOTFR	9.62	5.95	13.00	7.32	−1.48	0.15
TOTGR	7.23	5.07	9.81	5.78	−1.403	0.17

IMPFR = Pulses, frequency; IMPGR = Impulses, gravity; ORGFR = Organic, frequency; ORGGR = Organic, gravity; ANSFR = Anxiety, frequency; ANSGR = Anxiety, severity; HUMFR = Mood, frequency; HUMGR = Smoke, gravity; MANFR = Mania, frequency; MANGR = Mania, severity; AUTFR = Autism, frequency; AUTGR = Autism, severity; ESQFR = Schizophrenia, frequency; ESQGR = Schizophrenia, severity; ESTFR = Stereotypies, frequency; ESTGR = Stereotypies, severity; AGRFR = Self-harm, frequency; AGRGR = Self-harm, gravity; PSUFR = Sleep problems; frequency; PSUGR = Sleep problems, severity; TOTFR = Total frequency; TOTGR = Total severity. * *p* is significant < 0.05.

**Table 10 genes-14-01628-t010:** Pearson correlations between loss size and cognitive outcomes.

	Loss	Personal	Adaptative	Gross Mot.	Fine Mot.	Motor	Lang. Recep	Lang. Exp	Lan-Guaje	Cogn.	Total
Loss	-	−0.423 **	−0.467 **	−0.447 **	−0.451 **	−0.492 **	−0.490 **	−0.540 **	−0.537 **	−0.494 **	−0.491 **

** The correlation is significant at the level of 0.01 (bilateral). Mot., motor; Lang., language; Cogn., cognitive.

**Table 11 genes-14-01628-t011:** Pearson correlations between loss size and results of the BPI-01 subscales and totals.

	Loss	AUTFR	AUTGR	ESTFR	ESTGR	AGRFR	AGRGR	TOTFR	TOTGR
Loss	-	0.300 *	0.322 *	0.243	0.221	0.272	0.251	0.321 *	0.319 *

AUTFR = Self-harm behavior, frequency; AUTGR = Self-harm behavior, severity; ESTFR = Stereotyped behavior, frequency; ESTGR = Stereotyped behavior, severity; AGRFR = Aggressive behavior, frequency; AGRGR = Aggressive behavior, severity; TOTFR: Total frequency; TOTGR = Total severity. * The correlation is significant at the level of 0.05 (bilateral).

## Data Availability

Data are deposited in the publicly available database DECIPHER (https://www.deciphergenomics.org/, accessed 10 March 2021) included among the following accession numbers: 436269 to 436336 corresponding to cases 5pIMG01–5pIMG70.

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
