# Peer review of "Cognitive–Behavioral Profile in Pediatric Patients with Syndrome 5p-; Genotype–Phenotype Correlationships"

_genes, 2023, doi:10.3390/genes14081628_

Round 1
Reviewer 1 Report
This article presents a groundbreaking study that delves into the cognitive-behavioral profile of children with 5p minus syndrome (S5p-). The research addresses the pressing need for a comprehensive understanding of this neurodevelopmental disorder, which is caused by the deletion of the short arm of chromosome 5. By exploring various aspects such as biomedical, genetic, cognitive, and behavioral factors, the study takes a multidisciplinary approach, providing valuable insights into the syndrome.
One of the strengths of this research lies in its thorough methodology. The selection of appropriate instruments for neuropsychological assessment ensures the accuracy and reliability of the findings. By employing a cohort study design, the researchers were able to analyze a significant number of children with S5p-, contributing to the robustness and generalizability of the results.
The study sheds light on several key findings. Notably, it highlights that children with S5p- generally exhibit higher cognitive levels compared to their communicative and motor abilities. This nuanced understanding challenges previous assumptions and enriches our knowledge of the syndrome. The identification of language difficulties, particularly expressive language, as a crucial factor influencing the frequency and severity of behavioral problems is an important contribution to the field. Moreover, the correlation between the size of genetic material loss on 5p and the cognitive level of the subjects offers valuable insights into the underlying mechanisms of the syndrome.
The authors' conclusions provide valuable implications for clinical practice. Recognizing that language, particularly expressive language, significantly modulates the behavioral aspects of individuals with S5p-, the study emphasizes the importance of tailored verbal or alternative communication strategies. This crucial recommendation can have a positive impact on the lives of those affected, improving their overall well-being and quality of life.
Overall, this article represents a significant step forward in our understanding of 5p minus syndrome. Its interdisciplinary approach, meticulous methodology, and valuable findings contribute to the scientific community's knowledge base. The study has the potential to shape future research, clinical interventions, and support strategies for individuals with S5p- and their families.
Author Response
We want to acknowledge to Referee 1 for such comments and because he/she got the essential idea of this work; the study has the potential to shape future research, clinical interventions, and support strategies for individuals with S5p- and their families.
Reviewer 2 Report
The manuscript untitled "Cognitive-behavioral profile in pediatric patients with syndrome 5p-. Genotype-phenotype correlationships." is novel because of the lack of investigation of this syndrome. The text is globally well-written and accessible. Some points can be improved and/or precised:
- line 49: delete the point to continue the sentence "... variability which is related..."
- line 49: add a reference for this idea concerning the variability due to genetic material
- line 57 and next: precise the ages of the participants. For this study, the authors focused the results on IQ (so the WISC evaluation), so is it pertinent to mention all the other tests? or say a word on these results? It is unclear for me.
- line 79 and next: what tests or questionnaires are used to investigate behavioral, attentional problems?
- line 87: replace "characteristics" by "symptoms"
- line 103: the age range is very large ; what sense? I understand the difficulty of recruitment but some children do not have access to language, some are at school. It could be taken in account for the results.
- presentation of tests: it should more justified because of the age of the population. Regarding the studied population and the population for whom the tests are done, it is questionable. An argumentation is needed: some tests are for younger children and some for children and adults...
- line 268: "Tests", delete the upper case
- line 273: "The mean age", delete the upper case
- line 333: "mean age of mean age" should be corrected
- line 464: section 3.7. should be in italics
- line 488: section 3.8. should be presented differently, it is hard to follow; the presentation should be changed (a table?)
- line 737: conclusions; the authors indicate that communicative and motor aspects should be at the first level of intervention. I think this needs to be modulated with regard to age, to take greater account of the developmental dimension.
Author Response
We want to acknowledge to Referee 2 for all comments and helping us to improve the quality of this MS.
The manuscript untitled "Cognitive-behavioral profile in pediatric patients with syndrome 5p-. Genotype-phenotype correlationships." is novel because of the lack of investigation of this syndrome. The text is globally well-written and accessible. Some points can be improved and/or precised:
- line 49: delete the point to continue the sentence "... variability which is related..."
.- Accepted, the sentence continues now
- line 49: add a reference for this idea concerning the variability due to genetic material
Accepted, a reference has been added, based in our previous experience Nevado et al., 2021. Deep Phenotyping and Genetic Characterization of a Cohort of 70 Individuals With 5p Minus Syndrome
- line 57 and next: precise the ages of the participants. For this study, the authors focused the results on IQ (so the WISC evaluation), so is it pertinent to mention all the other tests? or say a word on these results? It is unclear for me.
Accepted, a new sentence including age of the participants has been included: They analyzed the scores of 26 subjects with age between 6 years and 4 months to 15 years and 5 months (mean 8 years and 3 months)
Regarding the second part of the question we highlighted only result from WISC because we are focus in cognitive data in that paragraph: Among the cognitive results it is remarkable that, the children obtained an IQ average score of 47.81. On the verbal scale, the mean obtained scores was 50.3. They found no significant differences between the verbal and manipulative scales. The study also showed a higher prevalence of people with moderate than severe intellectual disability.
- line 79 and next: what tests or questionnaires are used to investigate behavioral, attentional problems?
Accepted, we include a couple of references regarding test used in this aspect requiring the referee 2:
In terms of behavioral aspects, individuals with S5p- are very curious about novelty, interested in what is happening around them and others. The most relevant characteristics of their character and behavior are: to have a marked sense of humor, to be affectionate, scary, and shy. Dikens y Clarke used The Aberrant Behaviour Checklist Aman, M. G., Burrow, W., & Wolford, P. L. (1995). The Aberrant Behavior Checklist Community: Factor validity and effect of subject variables for adults in group homes. American Journal on Mental Retardation, 100, 283–292.) to 146 individuals with age between 2 and 40 years (mean12 yrs), observing that, they rarely present withdrawal or psychotic behavior (22). Although, different works and investigations have highlighted that people with S5p-, presented important and varied of behavioral problems (15, 22-29). Wilkins and colleagues evaluated motor development aspects of 65 individuals with age between 6 ans 85 years byusing the Vineland Social Maturity Scale (Doll, E. A. Vineland (N.J.). Training School y American Guidance Service (1965). Vineland social maturity scale: Condensed manual of directions. Ed. American Guidance Service, Circle Pines, Minn) observing that most of these studies have focused on attention and hyperactivity problems, present in more than half of the children evaluated (30), but there is little research on self- and hetero-aggressive behaviors (16).
- line 87: replace "characteristics" by "symptoms"
Accepted, “symptoms” replaced “characteristic” in this paragraph, one term more appropriate, thanks.
- line 103: the age range is very large ; what sense? I understand the difficulty of recruitment but some children do not have access to language, some are at school. It could be taken in account for the results.
It is a reality, as seen in all the studies carried out, the age ranges are wide since, being a rare disease, and it is difficult to recruit individuals. It is also true that this has to be taken into account when analyzing the results, mainly when analyzing the results of motor and language development. In addition, the selection of measure tests in individuals with ID is a challenge since this evaluation must occur out of range or tests are used whose scales and standardized scores do not adjust to the population studied.
- presentation of tests: it should more justified because of the age of the population. Regarding the studied population and the population for whom the tests are done, it is questionable. An argumentation is needed: some tests are for younger children and some for children and adults...
We explain this aspect into this paragraph: Neuropsychological assessment instruments.
The selection of instruments to measure cognitive and behavioral aspects was made taking into account: i) The characteristics of the participating subjects. The age of the minors, the presence of intellectual disability and language and motor difficulties. ii) The instruments used in published research, both with subjects with S5p- and with people with ID and characteristics similar to our participants. iii) The evaluation procedure. It is remarkable that the difficulties derived from evaluating subjects with intellectual disabilities mean that, in many cases, this evaluation must occur out of range or tests are used whose scales and standardized scores do not adjust to the population studied
- line 268: "Tests", delete the upper case
Accepted.
line 273: "The mean age", delete the upper case
- Accepted.
- line 333: "mean age of mean age" should be corrected
-- Accepted.
- line 464: section 3.7. should be in italics
- Accepted.
- line 488: section 3.8. should be presented differently, it is hard to follow; the presentation should be changed (a table?)
- Accepted, we revised this large paragraph trying to generate a more visual reading for the reader including a new figure, the figure 5. Statically data were also in three new tables at supplemental data (Tables 7-9).
- line 737: conclusions; the authors indicate that communicative and motor aspects should be at the first level of intervention. I think this needs to be modulated with regard to age, to take greater account of the developmental dimension.
- Accepted. We modulated that sentence, including corrected by age: This implies that these two aspects corrected by age, must be two fundamental milestones in a first level of intervention.
Reviewer 3 Report
In this manuscript, the authors described the developmental profile of a cohort of 45 children with 5p minus Syndrome (S5p-), considering biomedical, genetic, cognitive, and behavioral aspects. Notably, they found that self-harm was the most significant behavioral issue, particularly among girls. Compulsive behavior, limited preferences and interest in monotony are significantly more frequent in subjects with higher cognitive levels. Additionally, they also found a significant correlation between the size of the genetic material loss on chromosome 5 and the cognitive level of the affected individuals. This study established potential genotype-phenotype relationships within the cohort.
I appreciate the authors' efforts in describing the developmental profile of the individuals with S5p- and identifying significant behavioral and cognitive characteristics. However, there are some areas that need improvement:
1. Lack of Reference Information: The manuscript lacks reference information for each assessment, and there is no control group provided for comparison. It is essential to include normal ranges or control group data for better interpretation of the scores in each test. This will help readers understand the severity of the phenotype observed in the cohort compared to typical development.
2. Data Presentation: Figures 2 and 3 show the distributions of motor aspects and language-communicative aspects of these individuals, but the actual age of each subject is missing. Without this information, it becomes challenging for readers to gauge the severity of the phenotype at different ages. Including the age of each individual in the figures would enhance the clarity and usefulness of the data.
Perhaps part of the information is provided elsewhere in the manuscript or people can refer to relavant publications, but it is difficult to read and quickly understand without the same information shown in a chart or table. Addressing these comments will strengthen the manuscript and provide a more comprehensive understanding of the research findings.
Need minor edits, eg: please pay attention to the single and plural forms.
Author Response
We want to acknowledge to Referee 3 for all comments and helping us to improve the quality of this MS.
In this manuscript, the authors described the developmental profile of a cohort of 45 children with 5p minus Syndrome (S5p-), considering biomedical, genetic, cognitive, and behavioral aspects. Notably, they found that self-harm was the most significant behavioral issue, particularly among girls. Compulsive behavior, limited preferences and interest in monotony are significantly more frequent in subjects with higher cognitive levels. Additionally, they also found a significant correlation between the size of the genetic material loss on chromosome 5 and the cognitive level of the affected individuals. This study established potential genotype-phenotype relationships within the cohort.
I appreciate the authors' efforts in describing the developmental profile of the individuals with S5p- and identifying significant behavioral and cognitive characteristics. However, there are some areas that need improvement:
- Lack of Reference Information: The manuscript lacks reference information for each assessment, and there is no control group provided for comparison. It is essential to include normal ranges or control group data for better interpretation of the scores in each test. This will help readers understand the severity of the phenotype observed in the cohort compared to typical development.
First of all, it has to be remarked that The Battelle Developmental Inventory is a battery designed to assess the key developmental skills of children from birth to age 8. Although, some of the subjects in the sample were over the chronological age of the one collected in this test, none of them had a higher developmental age. In fact, globally their mental age is around 1.72 years old with SD of 1.32 years (see Table 2), whereas mean chronological age is around 6 years. Thus, all the individuals were two or more standard deviations below their chronological age. Using the Battelle booklet, we can use direct scores to translate into chronological age, IQ and typical scores. We thought that the best way to perform the analysis of the subjects, both individually and in groups, was to use the developmental age, since the rest of the scores were so low in relation to their reference group that they would not allow comparisons. Thus, in order to delve into the most relevant cognitive aspects of the population with S5p, the levels obtained in the motor and language areas were analyzed.
Regarding BPI-01 test we remark that this test has been developed over the years and has undergone several revisions. It was initially developed to evaluate adults with ID. It has since been revised and modified and used as an assessment instrument for children, adolescents and adults (Rojahn et al., 2001). It is completed by the parents, or a person close to the subject, who must assess the frequency and severity of the behaviors described. The evaluation is made through a Likert scale of 4 points for the frequency of behavior (every month, every week, daily, every hour), and 3 points for severity (mild, moderate, severe). If the behavior does not occur, 0 is scored. So, it is completely difficult to assess the normality to be compared for that reason that an individual without ID or behavioral problems does not score at all. Thus, higher score indicate major behavior problems into a questionnaire consisting of 52 items that measures the self-injurious behavior (14 items), stereotyped behavior (24 items) and aggressive/destructive behavior (11 items) of the subjects evaluated. We modified the table5 for improving its comprehension. Table 5 presents the data obtained in the behavioral rating scale of BPI-01. Forty subjects scored on one of the items of the three subscales that make up the BPI-01: self-harm behavior, stereotyped behavior and aggressive/destructive behavior, with the most frequent behaviors being self-harm (N = 37), with the highest mean score (mean= 4.6; SD=3.79). In each of the three subscales, two types of scores were obtained. The frequency with which the behavior has appeared in the past 12 months (Frequency) and the severity of the behavior (Severity).
Regarding The Repetitive Behaviours Questionnaire (RBQ) offers a clinical cut-off point for the different elements of the subscales. Thus behaviors that occur "once a day" or "more than once a day" were considered clinically relevant, that is, if a score of three or four points is obtained in this item. An individual without ID nor behavioral problems does not score more than 1 point. Table 6 presents the results of the Repetitive Behavior Questionnaire (RBQ). The five subscales of the test: stereotyped behavior, compulsive behavior, limited preferences, repetitive speech and insistence on monotony, collect the frequency of the behavior described in the last month. Thirty-nine children scored on some of the items on the scale, mainly on limited preferences (N = 18) and insistence on monotony (N = 18). None of the individuals scored at repetitive speech, which seems not to be a core item for behaviour issues in the syndrome.
The Diagnostic Evaluation for the Severely Disabled (DASH-II) is composed of 84 items that allow detecting psychiatric and emotional disorders in adults with ID and great support needs. Different authors have used it to also evaluate minors (Paclawskyj et al., 1997; Valdovinos et al., 2004). The informant must know the person evaluated for at least six months and scores the frequency, duration and severity of the behaviors described in each item. In our research work, we decided to analyze the scores obtained from the frequency and severity of the items evaluated, since the analysis of the duration of the period in which these behaviors are occurring is not significant, due to the young age of the participants. The results of the subarea -excretion disorders- (since most children do not control sphincters), and those of eating disorders and sexual behavior disorders (little Table 7 shows the results of the Diagnostic Evaluation for the Severely Disabled (DASH-II). The subjects who score in any of the subscales used in the test: impulses, organic, anxiety, mood, mania, autism spectrum disorder, schizophrenia, stereotypies, self-injurious behavior and sleep problems, are forty-four. However, none reaches the necessary score to be able to establish any of the diagnoses of the scale, whose score, as we have seen, is obtained through a Liker-type scale. The breadth of the ranges of scores achieved in each of the subscales is small, between 0-1 (organic scale/severity) and 0-7 (autism scales). Thirty-six of the children have some of the traits that the scale recognizes as characteristic of autism, and thirty-four, self-injurious behaviors. Stereotyped behavior (N = 31), sleep problems (N = 28) and impulsive behavior (N = 24) are also common. Some score in organic disorder (N = 3), mood disorder (N = 16) and mania (N = 20), and none in anxiety or schizophrenia. Psychopathology should be ruled out in scores below (Matson et al., 1994): 8pts for impulses; 4-5 points for organic; 2 points for anxiety; 6 points for Mood aspects; 4-5points in mania; 4-5points in autism, 2-3 points in schizophrenia; 4-5 points in stereotypes; >1.5 pts for self-harm injured or sleeping problems (based in in our professional experience)
- Data Presentation: Figures 2 and 3 show the distributions of motor aspects and language-communicative aspects of these individuals, but the actual age of each subject is missing. Without this information, it becomes challenging for readers to gauge the severity of the phenotype at different ages. Including the age of each individual in the figures would enhance the clarity and usefulness of the data.
It is true, Figure 2 needs to be supported by Table 6 of supplemental data, in which we put the motor milestones in relation to the age of the individuals. Figure 3 also need to be supported by using Tables 3 and 4.
Perhaps part of the information is provided elsewhere in the manuscript or people can refer to relevant publications, but it is difficult to read and quickly understand without the same information shown in a chart or table. Addressing these comments will strengthen the manuscript and provide a more comprehensive understanding of the research findings.
Comments on the Quality of English Language. Need minor edits, eg: please pay attention to the single and plural forms.
-Accepted, the MS has been revised for English editing.